# Olefin oligomerization by main group Ga$^{3+}$ and Zn$^{2+}$ single site catalysts on SiO$_2$

Nicole J. LiBretto [1,3], Yinan Xu [1,3], Aubrey Quigley[1], Ethan Edwards [1], Rhea Nargund[1], Juan Carlos Vega-Vila[1], Richard Caulkins[1], Arunima Saxena[1], Rajamani Gounder [1], Jeffrey Greeley[1], Guanghui Zhang [1,2✉] & Jeffrey T. Miller [1✉]

In heterogeneous catalysis, olefin oligomerization is typically performed on immobilized transition metal ions, such as Ni$^{2+}$ and Cr$^{3+}$. Here we report that silica-supported, single site catalysts containing immobilized, main group Zn$^{2+}$ and Ga$^{3+}$ ion sites catalyze ethylene and propylene oligomerization to an equilibrium distribution of linear olefins with rates similar to that of Ni$^{2+}$. The molecular weight distribution of products formed on Zn$^{2+}$ is similar to Ni$^{2+}$, while Ga$^{3+}$ forms higher molecular weight olefins. In situ spectroscopic and computational studies suggest that oligomerization unexpectedly occurs by the Cossee-Arlman mechanism via metal hydride and metal alkyl intermediates formed during olefin insertion and β-hydride elimination elementary steps. Initiation of the catalytic cycle is proposed to occur by heterolytic C-H dissociation of ethylene, which occurs at about 250 °C where oligomerization is catalytically relevant. This work illuminates new chemistry for main group metal catalysts with potential for development of new oligomerization processes.

---

[1] Davidson School of Chemical Engineering, Purdue University, West Lafayette, IN, USA. [2] State Key Laboratory of Fine Chemicals, PSU-DUT Joint Center for Energy Research, School of Chemical Engineering, Dalian University of Technology, Dalian, Liaoning, PR China. [3] These authors contributed equally: Nicole J. LiBretto, Yinan Xu. ✉email: gzhang@dlut.edu.cn; mill1194@purdue.edu

Olefin oligomerization to produce higher molecular weight olefins was commercialized in the 1960s and often utilizes homogeneous, transition metal catalysts containing $Cr^{3+}$ and $Ni^{2+}$[1–3]. Commercial processes utilizing Ni-based homogeneous catalysts include the Shell Higher Olefin Process (SHOP), SABIC/Linde alpha-SABLIN, DuPont's Versipol™, IFP AlphaSelect™, and UOP Linear-1™ processes[1,4,5]. Chevron Philips (Gulf) and others utilize similar catalysts under similar reaction conditions[6]. Such homogeneous catalysts also often require aluminum–alkyl co-catalysts to form the initial metal–alkyl reaction intermediate. In order to obtain a high selectivity for linear α-olefins, mild reaction temperatures ranging from 30 to 170 °C are generally used[7–9]. High pressures, often >25 atm, are required to obtain high olefin conversion[10]. Homogeneous $Ni^{2+}$ catalysts offer high oligomerization rates, and the selectivity is often tailored to favor low molecular weight, linear α-olefins, useful for polymer applications[4,10,11]. Separation and regeneration of homogenous catalysts is generally not possible; thus, there is interest in development of heterogeneous catalysts, which are readily separated from the products and can be regenerated. The latter include immobilized $Ni^{2+}$ sites on zeolite (BEA, MFI), or mesoporous aluminosilicates (MCM-41) and other high surface area oxide supports which sometimes generate Brønsted acid sites during the reaction, leading to bifunctional catalysis[10,12–17]. Supported, heterogeneous oligomerization catalysts typically do not require an aluminum alkyl co-catalyst and have lower activity, generally requiring higher operating temperatures than homogeneous catalysts.

For homogeneous and heterogeneous Ni-based catalysts, the Cossee–Arlman reaction mechanism (Fig. 1) is generally accepted. Ni-alkyl and hydrides are proposed key reaction intermediates, and olefin insertion and β-hydride elimination are the key elementary reaction steps, though limited spectroscopic evidence exists[18,19]. Activation and initiation of the catalytic cycle often occurs by alkyl transfer from an Al-alkyl co-catalyst to the Ni complex. Chain growth occurs by olefin insertion. β-hydride elimination of the longer metal alkyl leads to the olefin products and the formation of a metal hydride intermediate. Ethylene insertion to the metal hydride regenerates the metal alkyl intermediate completing the catalytic cycle[12,13]. It is generally accepted that empty *d* orbitals of the transition metal catalysts are required for olefin coordination-insertion and β-hydride elimination and the formation of metal alkyl and metal hydride reaction intermediates[20].

Recently, silica-supported, single site $Ga^{3+}$ and $Zn^{2+}$ (oxides) were reported for alkane dehydrogenation and olefin hydrogenation where metal hydride and alkyl intermediates and olefin insertion and β-hydride elimination elementary steps were proposed[21–25]. Based on the common reaction intermediates and elementary steps, we hypothesize that there is a mechanistic relationship between dehydrogenation, hydrogenation, and oligomerization reaction pathways and in this manuscript show that single site $Ga^{3+}$ and $Zn^{2+}$ also catalyze olefin oligomerization. The spectroscopic and theoretical evidence demonstrate the formation of reaction intermediates and elementary steps characteristic of transition metal Cossee–Arlman oligomerization mechanism also occurs on these main group metal ion sites.

## Results

**Initial catalyst structure.** The initial $Ga/SiO_2$ and $Zn/SiO_2$ precatalyst was synthesized following previously reported methods[22,23], and the structure was determined by in situ X-ray absorption spectroscopy (XAS), including both XANES and EXAFS (Supplementary Table 1) and compared to that of $Ni/SiO_2$. The catalysts were dehydrated at 550 °C in He and compared to known reference compounds at each metal edge (Supplementary Fig. 1). $Ga/SiO_2$ and $Zn/SiO_2$ have a similar structure to $Ni/SiO_2$, in that each has 4 M–O bonds with no evidence of M–O–M higher shell scattering (M = Ni, Ga, Zn). These results are consistent with previous single site structures reported for $Ga^{3+}$, $Zn^{2+}$, and $Ni^{2+}$ hydrogenation and dehydrogenation catalysts[22,23,26,27]. These structures differ from organometallic catalysts where metal alkyls, like trimethylgallium, were grafted to a $SiO_2$ support resulting in M–M bonds[28–30]. The rates and selectivities of single site $Ga/SiO_2$ and $Zn/SiO_2$ for propane dehydrogenation and propylene hydrogenation are similar to those previously reported (Supplementary Tables 2 and 3, respectively)[23,31]. Taken together, the XAS structures and catalytic performance confirm that single site, main group $Zn^{2+}$ and $Ga^{3+}$ silica supported catalysts have been prepared in agreement to previous literature.

**Olefin oligomerization.** At 1 atm by varying $C_2H_4$ space velocity, up to 5% conversion was obtained at 250 °C with stable performance for at least 25 h for both $Ga/SiO_2$ and $Zn/SiO_2$ (Supplementary Fig. 2). The reaction rate was calculated by normalizing the rate of butene formation by the total moles of metal and

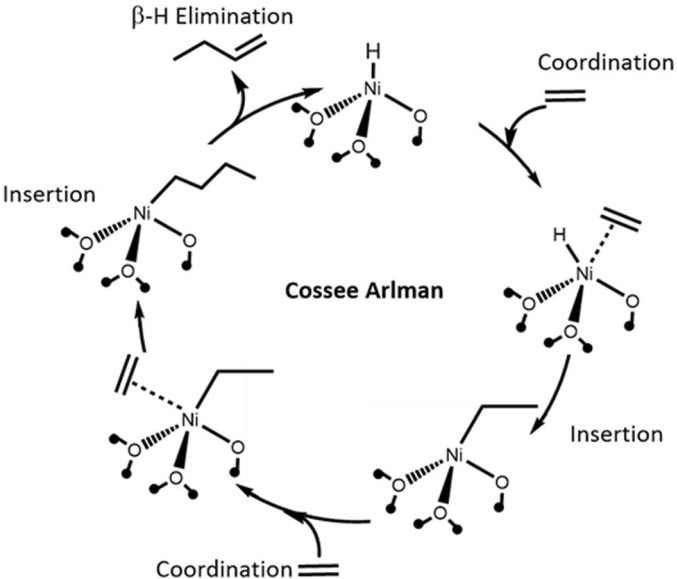

**Fig. 1 Cossee–Arlman Ni oligomerization mechanism.** Elementary reaction steps and intermediates[68].

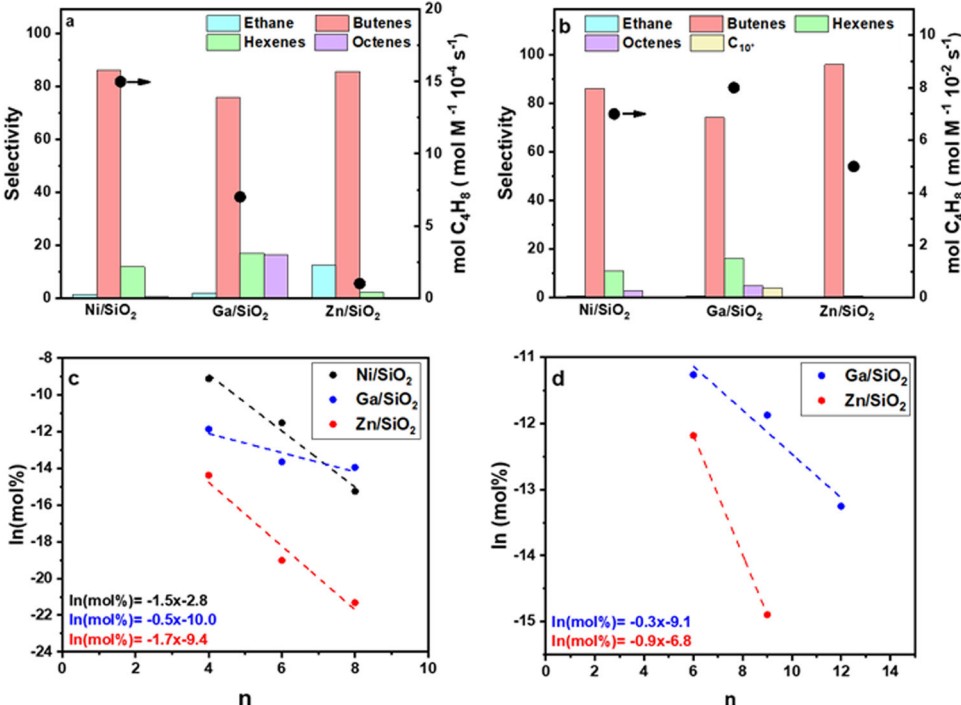

**Fig. 2 Ethylene oligomerization product distribution and reaction rate at 250 °C. a** 1 atm ($X$ = ~5%). **b** 30.6 atm ($X$ = ~20%). **c** $C_2H_4$ Schultz Flory distribution. **d** $C_3H_6$ Schultz Flory distribution.

increased in the order of $Ni^{2+} > Ga^{3+} > Zn^{2+}$ (Fig. 2a, Supplementary Table 4). $Ni^{2+}$ is known for its high selectivity to dimers (i.e. butenes). On $Ni/SiO_2$, at 5% conversion, there is a high selectivity towards butenes ($C_4^=$, 86%) with lower selectivity towards hexenes ($C_6^=$, 12%). Trace octenes ($C_8^=$) and higher molecular weight products are formed, along with a small amount of ethane ($C_2H_6$, <2%).

For $Ga/SiO_2$ at the same conversion (5%), the $C_4^=$ selectivity (76%) is lower and the $C_6^=$ (17%) and $C_8^=$ (6%) selectivities are higher than those observed for $Ni/SiO_2$. $Ga/SiO_2$ has a comparable reaction rate to $Ni/SiO_2$. Single site $Zn/SiO_2$ also has a similar selectivity to $C_4^=$ (86%) as $Ni/SiO_2$[12]. The reaction rate (per metal) of $Zn/SiO_2$, however, is three times lower than that of $Ni/SiO_2$ and $Ga/SiO_2$. The $C_2H_6$ selectivity of $Zn/SiO_2$ is also higher (~12%) than on $Ni/SiO_2$ and $Ga/SiO_2$. At low conversion, oligomerization leads to formation of $C_4^=$ with an equilibrium distribution of 1-butene, *cis*-2-butene, and *trans*-2-butene. No isobutene ($iC_4^=$) was observed on any of these catalysts, indicating that there is no skeletal isomerization and only olefin isomerization occurs. Higher molecular weight products were also an equilibrium distribution of linear olefins. For each catalyst, there were many isomers of linear olefins. For example, there were five $C_6^=$ isomer products representing 1-hexene *cis* and *trans*-2-hexene, and cis and *trans*-3-hexene. Similarly, there were many isomer products for the $C_8^=$ and other products.

Higher $C_2H_4$ conversion can be achieved at higher reaction pressures. In addition, higher molecular weight olefins are also formed (Fig. 2b). For example, at 250 °C and 30.6 atm $C_2H_4$, conversions up to 20%, and rates two orders of magnitude higher than those at atmospheric pressures, were obtained (Fig. 2b, Supplementary Table 5). The results in Supplementary Table 5 include only the quantification of gas-phase products, which accounted for ~70–80% of all products. At higher rates, the selectivity toward ethane on all catalysts decreased to <0.5%, as higher conversion of $C_2H_4$ favored oligomerization over hydrogenation (Supplementary Table 5). Liquid products were collected

continuously during the reaction and were analyzed offline by mass spectrometry (GC–MS). On $Ga/SiO_2$, the liquid phase products showed at least a small concentration of linear hydrocarbons were formed up to $C_{15}$ (Supplementary Fig. 3). Additional products included small amounts of paraffins and saturated rings, but there was little evidence of branched olefins. This distribution is consistent with the non-acidic nature of the inert $SiO_2$ support.

The product distributions on each catalyst are consistent with a Schulz Flory distribution, which is determined by the ratio of the rate of olefin insertion, or propagation ($\alpha$), to β-hydride elimination, or termination ($1-\alpha$), and the Schultz Flory coefficient can be determined from the slope of the ln(mass fraction) verses ($2n$) carbon number to estimate the tendency to produce higher molecular weight products[32]. The molecular weight distribution for these catalysts for $C_2H_4$ oligomerization are shown in Fig. 2c. The Schultz Flory coefficient for $C_2H_4$ oligomerization for $Ga/SiO_2$ ($\alpha_{Ga}$) is 0.59, while that of $Zn/SiO_2$ ($\alpha_{Zn}$) is 0.18, and $Ni/SiO_2$ ($\alpha_{Ni}$) is 0.22. $Ni^{2+}$ is known for olefin dimerization and reported $\alpha$ values are between 0.2 and 0.3[12,33]. $Zn^{2+}$, like $Ni^{2+}$, favors low molecular weight products like butenes, while $Ga^{3+}$ produces some higher molecular weight oligomers.

For comparison, $C_3H_6$ oligomerization was also performed at 1 atm and 250 °C with product selectivities and rates given in Supplementary Table 6. The propylene oligomerization rate is 2–4 times higher than that for ethylene, and the Schulz Flory coefficient (Fig. 2d) was slightly higher, for example, $Ga/SiO_2$ ($\alpha_{Ga}$ = 0.70) and $Zn/SiO_2$ ($\alpha_{Zn}$ = 0.40), which is consistent with the Schultz Flory trends observed for other transition metal oligomerization catalysts[34].

**Evidence for oligomerization intermediates.** $Ga^{3+}$ and $Zn^{2+}$ catalysts have been proposed to heterolytically dissociate $H_2$ and C–H bonds, for example, during olefin hydrogenation and alkane

dehydrogenation, respectively. Heterolytic dissociation of $H_2$ on $Ga^{3+}$ and $Zn^{2+}$, or other single site catalysts, forms a M–H bond, i.e., a hydride ($H^−$), and a proton ($H^+$), which dissociates the M–OSi support bond[21,35,36]. Protonation of the bridging O atom leads to breaking of the M–O bond and formation of a new Si–OH group[23,26,31]. In situ spectroscopic studies for these reactions and intermediates, however, have generally been obtained on catalysts at much higher temperatures (>500 °C) than those for olefin oligomerization, ca. 250 °C. Here, we provide experimental evidence for the formation of the reaction intermediates and elementary steps at the latter, lower reaction temperatures. To demonstrate that these catalysts can perform the elementary steps required of the Cossee–Arlman mechanism, their ability to activate $H_2$, C–H bonds, and alkylate metal hydrides was evaluated.

Olefin hydrogenation and oligomerization occur at similar temperatures and are thought to form M–H intermediates. In situ XAS and IR spectroscopies demonstrate heterolytic dissociation of $H_2$ by single site $Ga^{3+}$ and $Zn^{2+}$ at temperatures where catalytic (hydrogenation) activity is observed. In $H_2$ with increasing temperatures from ambient temperature to 550 °C, there are continual changes in the shape of the Ga (Figs. 3a, 2c, Supplementary Fig. 4a) and Zn (Supplementary Fig. 4c) K-edge XANES. For Ga/SiO₂, the white line intensity decreases, and there is formation of a feature before the edge. Changes in XANES were

isolated by subtraction of the fresh, unreacted catalyst from that reacted with $H_2$ (Fig. 3b, Supplementary Fig. 4b, d), and are due to changes in the coordination geometry and energy of the vacant $p$-orbitals of the main group metal ion[25]. The magnitude of the $k^2$-weighted EXAFS of the catalysts treated in $H_2$ at elevated temperatures (Fig. 3b, Supplementary Fig. 5) contains a peak at about 1.5 Å (phase-uncorrected distance) due to Ga–O or Zn–O bonds. With increasing temperature there is a decrease in the M–O coordination number for both catalysts. M–H bonds are not detected by EXAFS, and the loss of M–O bonds (Supplementary Table 7) has been suggested to reflect formation of M–H bonds[23,26,31]. There is a loss from 4 to ~3 M–O bonds from the fresh to $H_2$-treated catalysts, respectively. In addition, the absence of second shell M–O–M scattering in the EXAFS indicates that the single site structure is maintained at high temperature in $H_2$.

Further evidence for M–H intermediates was investigated by H/D exchange. For example, after treatment of the catalyst with $H_2$, the M–H was reacted with $D_2$ in a temperature-programmed surface reaction (TPSR). The normalized HD product corresponds to the fraction of metal hydride sites (Supplementary Table 8). For Ga/SiO₂, after $H_2$ treatment at 250 °C, about 0.65 mol HD/mol Ga was formed, which is very similar to the decrease in Ga–O coordination in $H_2$ observed by EXAFS (Fig. 3f, Supplementary Fig. 6, Supplementary Table 7). The shape of the TPSR profile with increasing reaction temperature is also

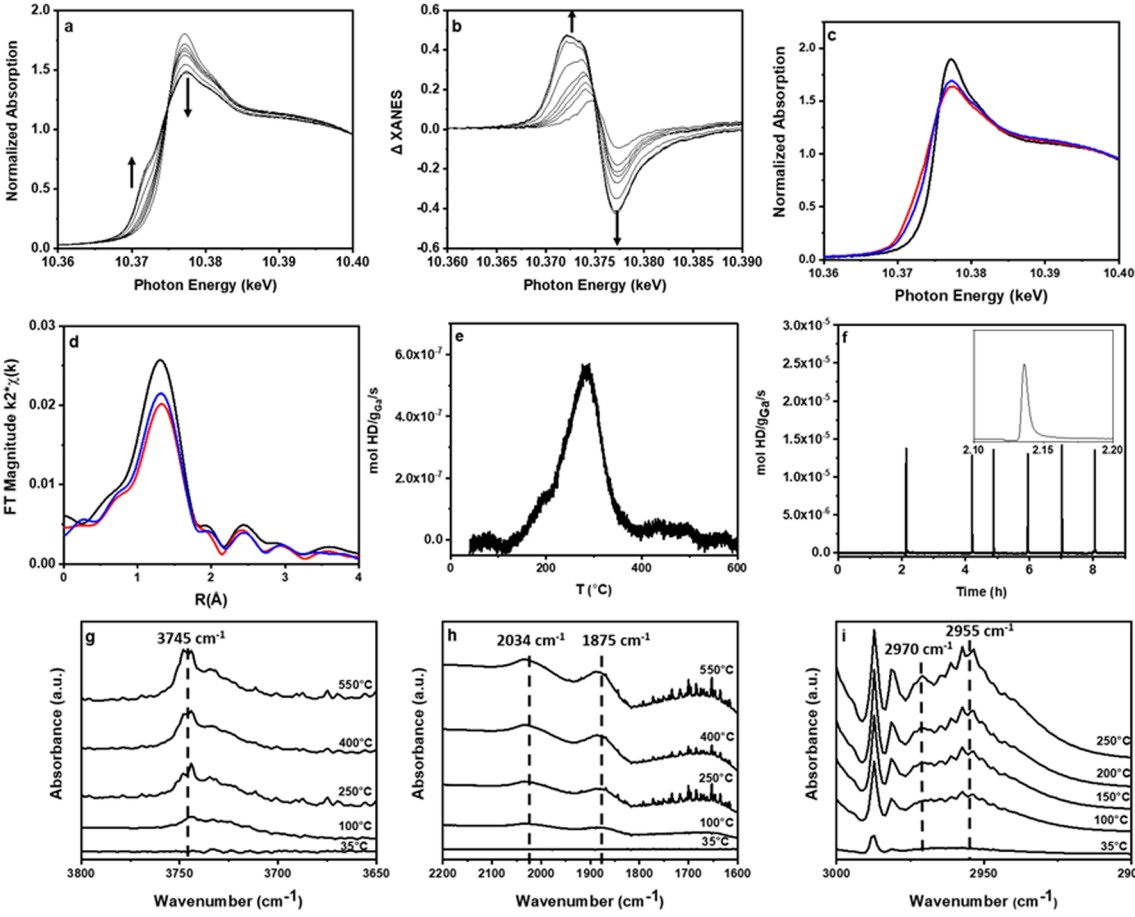

**Fig. 3 Characterization of reaction intermediates on Ga/SiO₂. a** Normalized Ga K edge XANES at increasing temperature from 35 to 550 °C in pure $H_2$. **b** The corresponding difference XANES. **c** Normalized Ga K edge XANES in $C_2H_4$ at 550 °C (blue), $H_2$ at 550 °C (red), and RT (black) sequentially. **d** The corresponding magnitude of the Fourier transform of the $k^2$-weighted EXAFS. **e** The temperature of HD desorption from M–H/$D_2$ isotopic exchange by TPSR, **f** time on stream for gaseous HD formation. **g** The difference IR Si–OH spectrum from 3800 to 3650 cm⁻¹ after treatment in pure $H_2$ at increasing temperature from 35 to 550 °C. **h** The corresponding difference IR Ga–H spectrum from 2200 to 1600 cm⁻¹. **i** The IR spectra in the C–H stretching region 2900-3000 cm⁻¹ after treatment in pure $C_2H_4$ at increasing temperature from 35 to 250 °C.

consistent with only one type of Ga species present (Fig. 3e, Supplementary Fig. 7). Similarly, $Zn/SiO_2$ forms 0.19 mol HD/ mol Zn at 250 °C, consistent with the changes in coordination number observed in Supplementary Table 7. Using these fractional surface coverages as a measure of the number of active sites at this temperature, the turnover rates (TOR) can be calculated and are similar for both catalysts (Supplementary Table 4). Higher reaction temperatures lead to larger amounts of H/D exchange, consistent with a larger number of M–H sites and higher rates (Supplementary Fig. 8, Supplementary Table 9).

In situ IR for $Ga/SiO_2$ also provides evidence for the heterolytic dissociation of $H_2$. The peak at 3745 $cm^{-1}$ corresponds to Si–OH vibrations. Reaction with $H_2$ at increasing temperature from 35 to 550 °C leads to an increased intensity in the Si–OH peak consistent with heterolytic $H_2$ dissociation at $Ga^{3+}$ sites. Figure 3g shows an increase in the intensity of the Si–OH peak with increasing temperature, which was determined by the difference of the spectrum at high temperature to that of the fresh catalyst before $H_2$ reaction.

In addition to an increase in the number of Si–OH, $H_2$ reaction on single site $Ga^{3+}$ also leads to two broad IR peaks at 2034 and 1875 $cm^{-1}$, which have been previously attributed to Ga–H stretching (Fig. 3h)[37–40]. As the temperature increases, the intensity of both peaks increases, consistent with the increase in number of Ga–H sites. The temperature dependence of the IR spectra of Si–OH and Ga–H indicate that $H_2$ dissociation occurs at temperatures as low as 200 °C, which is also the reaction temperature for olefin hydrogenation. Higher temperatures also favor the formation of an increased number of Ga–H sites, as indicated by the larger intensity of these IR bands.

The reactivity of the M–H with olefins, an elementary reaction of the olefin oligomerization pathway, was also demonstrated by in situ XAS. Following formation of Ga–H at 250 °C by reaction with $H_2$, the reacting gas was switched to $C_2H_4$, which led to additional changes in the XANES (Fig. 2a and Supplementary Fig. 6c). The EXAFS also shows a small increase in the first coordination shell, consistent with formation of new M–C bonds (Fig. 2b and Supplementary Fig. 6d). However, the XANES and EXAFS spectra are not identical to the original pre-catalyst suggesting that during the catalytic cycle, both metal hydride and metal alkyl intermediates contribute to the resulting spectra. These changes due to reaction of Ga–H with ethylene were isolated, and a difference analysis was performed to verify the number of new M–C bonds (Supplementary Fig. 9, Supplementary Table 10)[31,41–44]. For example, there are approximately 0.4 and 0.2 Ga–C and Zn–C bonds per metal ion, respectively, by alkylation of the hydrides. In addition, the first shell coordination number remains <4 after olefin alkylation suggesting that there are also M–H bonds present, ca. 0.4 and 0.5 per $Ga^{3+}$ and $Zn^{2+}$, respectively (Supplementary Table 7).

Ethylene insertion into the Ga–H was also confirmed by IR giving C–H stretching peaks from 2975 to 2940 $cm^{-1}$ (Supplementary Fig. 10). Thus, $C_2H_4$ insertion into Ga–H and formation of Ga-alkyl is a key elementary step and reaction intermediate characteristic of the Cossee–Arlman mechanism and readily occurs at the same temperature as oligomerization.

While reaction of $H_2$ confirms the formation of Ga–H, loss of Ga–O bonds, and formation of Si–OH, i.e., heterolytic dissociation, $H_2$ is not a reactant in olefin oligomerization. Thus, in situ IR of $C_2H_4$ on the $Ga/SiO_2$ pre-catalyst was obtained at temperatures from ambient to 250 °C and gives C–H stretching vibrations between 2975 and 2940 $cm^{-1}$ (Fig. 3i) with many of the same peaks as the Ga-alkyl spectrum (Supplementary Fig. 10). However, with ethylene there is an additional IR peak at 2970 $cm^{-1}$, which was not present in the Ga-alkyl spectra and has been previously assigned to a vinyl C–H stretch[45]. The IR spectra suggest that the heterolytic

C–H bond dissociation of $C_2H_4$ leads to formation of a Ga-vinyl intermediate at temperatures as low as about 100 °C. Higher concentrations are present at 250 °C, where the reaction rate is catalytically relevant. The formation of the M-vinyl intermediate has been proposed by DFT calculations as the initiation reaction in heterogeneous Ni oligomerization catalysts and is an additional elementary step in the heterogeneous oligomerization pathway, which is not required for homogeneous catalysts[13].

These spectroscopic results demonstrate the formation of reaction intermediates and elementary steps characteristic of the $Ni^{2+}$ Cossee–Arlman reaction mechanism, on these main group elements. Initiation of the catalytic cycle occurs by heterolytic dissociation of C–H bonds of ethylene. The elementary steps and reaction intermediates form at 250 °C where catalytic activity is observed, and increasing temperatures lead to higher concentrations of reaction intermediates.

**Oligomerization mechanism.** A periodic model for single site $Ga^{3+}$ ions on amorphous silica was created by the substitution of Si atoms by Ga. To maintain local charge balance after the substitution, a proton was added to an adjacent oxygen, resulting in an Si–OH moiety. Numerous representative sites, including both three-coordinated (3CN) and four-coordinated (4CN) Ga sites, were considered (Supplementary Figs. 11 and 12). 4CN sites exhibit shorter Ga–O bonds, with Si–OH near the active Ga site, and the four Ga–O bonds have an average bond length of 2.0 Å, in agreement with the pre-catalyst structure determined by XAS. Below, we focus on the catalytic properties of these sites. Modeling was done for the initiation steps of the single site $Ga/SiO_2$ pre-catalyst, leading to the first reaction intermediate in the catalytic cycle, i.e., Ga–H, followed by additional calculations to elucidate the catalytic pathway.

Initiation of the catalytic cycle begins with heterolytic dissociation of a C–H bond in $C_2H_4$ across one Ga–O bond. The initial catalyst structure (Fig. 4a) contains a bridging Si–OH, and a second non-bridging Si–OH is formed during the activation of $C_2H_4$. During this process, the vinyl carbon atom develops a negative charge. The initial ethylene C–H ($sp^2$) bond cleavage has an activation free energy of 1.69 eV. Migratory insertion of $C_2H_4$ into the resulting Ga-vinyl intermediate (1.47 eV) forms a Ga-butenyl intermediate. Subsequent β-H elimination (2.13 eV) leads to stoichiometric amounts of butadiene and a Ga–H intermediate (Supplementary Fig. 13, Supplementary Table 11). Since one butadiene molecule is formed per Ga site, butadiene was not detected experimentally, but the formation has been proposed based on DFT calculations for $Ni^{2+}$ heterogeneous oligomerization catalysts[13,14].

The energy landscape for catalytic oligomerization, beginning with the Ga–H intermediate generated using the 4CN Ga model, is illustrated in Fig. 4c, and the structures of the intermediates are given in Fig. 4d (Supplementary Fig. 14, Supplementary Table 12). $C_2H_4$ insertion into Ga–H yields a Ga-ethyl intermediate with an activation free energy of 1.82 eV. Subsequent insertion of a second $C_2H_4$ molecule yields a Ga–n-butyl intermediate with activation free energy of 2.06 eV. Finally, β-H elimination (2.03 eV) results in the desorption of olefin product, re-forming of Ga–H, and completion of the catalytic cycle. These energies are similar to those found in previous DFT calculations of $C_3H_8$ dehydrogenation for the β-H elimination of a propyl group on $Ga/SiO_2$ and $Zn/SiO_2$[23,46].

The oligomerization pathway was also calculated for a less constrained active site, starting from a Ga–H intermediate generated from a 3CN Ga site, which leads to slightly lower activation barriers (Supplementary Fig. 15, Supplementary Table 13). With the less-constrained geometry, the activation barrier of ethylene insertion on a 3CN site is slightly lower than that on a 4CN site. Similar site heterogeneity has been observed in the

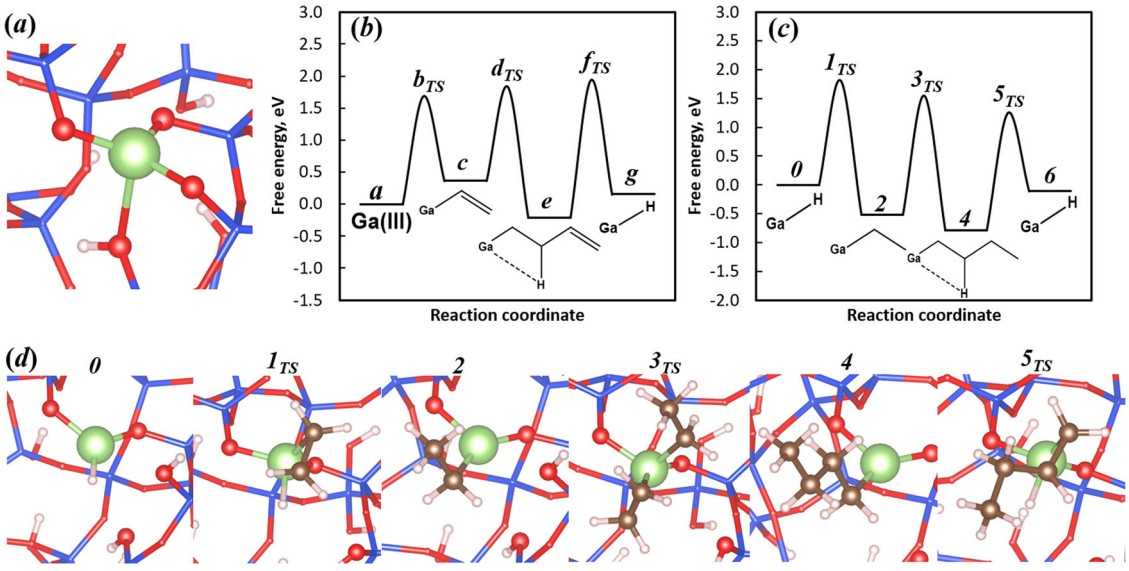

**Fig. 4 Ga structures and energies during activation and oligomerization catalysis. a** Ga site model where Ga = green, O = red, Si=blue, and H = white. **b** Free energy diagram of Ga–H formation on 4CN Ga site ($T$ = 250 °C). **c** Free energy diagram of oligomerization on Ga–H ($T$ = 250 °C). **d** Schematic of ethylene oligomerization on Ga–H and the associated reaction intermediates and transition states.

ethylene polymerization study using $Cr^{3+}$ sites, as well as propane dehydrogenation using $Ga^{3+}$, where similar amorphous silica models and site creation schemes were employed, suggesting that any such minority sites that may be present could also contribute to reactivity[46,47]. With the possibility of lowered barriers on less-constrained sites, the proposed mechanism (Fig. 5) is in line with the Cossee–Arlman scheme that was previously identified for $Ni^{2+}$ ions on zeolite supports[12,13].

## Discussion

This work demonstrates that isolated, $Ga^{3+}$ and $Zn^{2+}$ ion sites on inert $SiO_2$ support catalyze olefin oligomerization following the Cossee–Arlman mechanism which is generally accepted for transition metal catalysts. This chemistry, however, is unprecedented on main group metal ions. While previously propylene reacts with $Ga^{3+}$ ions on solid acid supports, such as MFI or ZSM-5 zeolite; however, the higher molecular weight product distribution is characteristic of carbenium ion catalysis, for example, skeletal isomerization and cracking, likely from active Brønsted acid sites. Here, the catalytic performance is due to exclusively $Zn^{2+}$ and $Ga^{3+}$ metal ion sites, which form linear olefins with even carbon numbers.

Further, this work suggests that at least for single site catalysts, olefin hydrogenation/alkane dehydrogenation and olefin oligomerization are mechanistically related through the formation of the same reaction intermediates and elementary steps. This suggests that any catalyst that is active for one reaction should also be active for the other two if they are structurally stable under all reaction conditions. The single site $Ni^{2+}$, in this study, performs oligomerization; however, $Ni/SiO_2$ reduces to metallic ($Ni^0$) nanoparticles at temperatures higher than about 350 °C, thus is not stable at the high temperature where alkane dehydrogenation is performed[48]. In addition, a recent study demonstrates that single site $Ni^{2+}$ catalysts in Zr-MOF NU1000 catalyze olefin oligomerization and hydrogenation[49,50]. Consistent with this proposal, single site $Cr^{3+}$ is catalytic for alkane dehydrogenation in the Catofin process, and olefin oligomerization, i.e., the Phillips catalyst[47,51–54]. Thus, there is evidence from heterogeneous

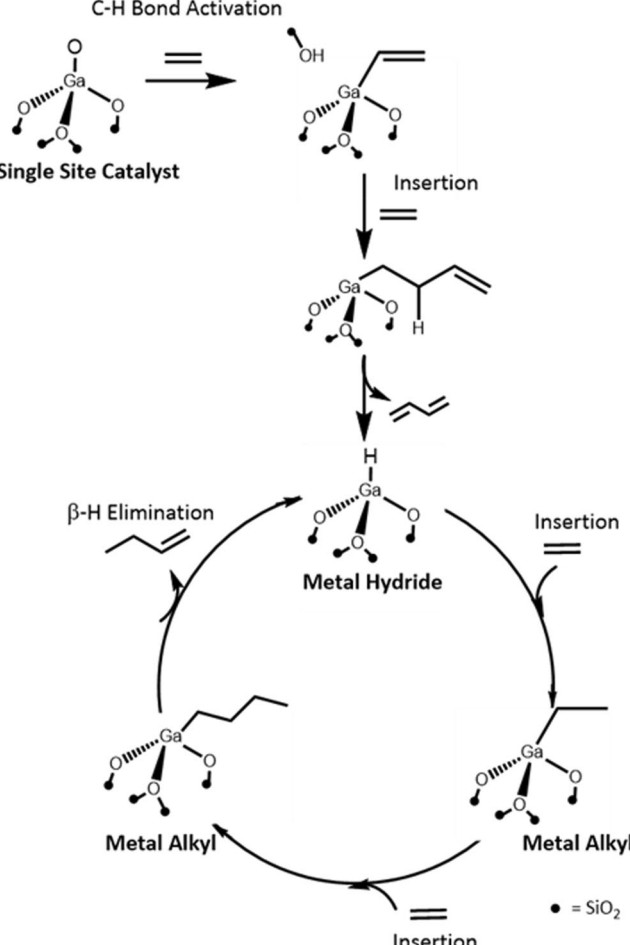

**Fig. 5 Proposed oligomerization pathway for single site, $Ga^{3+}$ catalysts.** Formation of the pre-catalyst to the first Ga–H intermediate, elementary steps and reaction intermediates.

transition metal catalysts that all three reactions are mechanistically related.

Based on mechanistic and computational studies of Ni oligomerization catalysts, it is unexpected that main group single site catalysts would have similar chemistries. For example, for Ni, C–C and C–H bond activation requires transfer of electron density from the $C_2H_4$ highest occupied molecular orbital to unfilled $3d$ orbitals, while, Ni $3d$ electron density is donated to the $C_2H_4$ lowest unoccupied molecular orbital, thus forming a π-bond[20,55]. For transition metal catalysts, therefore, the elementary steps (olefin insertion, C–H activation, and β-H elimination) generally involve transition state interactions and bond formation with $3d$ orbitals. However, main group metals, like $Ga^{3+}$ and $Zn^{2+}$, with $3d^{10}$ electron configurations, would not be expected to participate in these catalytic steps. Thus, the ability of single site $Ga^{3+}$ and $Zn^{2+}$ sites to catalyze oligomerization by the same reaction pathway as $Ni^{2+}$ is unexpected. Nevertheless, as this study shows, the same elementary steps and reaction intermediates are common for $Ni^{2+}$, $Ga^{3+}$, and $Zn^{2+}$. In addition, we suggest that hydrogenation/dehydrogenation (Supplementary Fig. 16) and oligomerization reaction pathways (Fig. 5) are related through the formation of the same reaction intermediates (M–H and M-alkyl) and elementary reaction steps (olefin coordination and insertion and β-H elimination). While the specific orbital interactions of these main group active sites with reactants and intermediates remains to be established, the absence of $3d$ valence orbitals suggests that valence $p$-orbitals may participate in the catalytic activation, contrary to transition metals where the metal-adsorbate $d$-orbital bond formation is essential for adsorption, activation, and reaction throughout the catalytic cycle. The results of this study also suggest that other catalytic reactions may be possible with main group elements.

In summary, main group $Ga^{3+}$ and $Zn^{2+}$ single site catalysts unexpectedly catalyze olefin oligomerization at temperatures as low as 250 °C. Initiation of the catalytic cycle begins by heterolytic dissociation of vinyl C–H bonds of ethylene, for example, and the reaction intermediates and elementary steps suggest a Cossee–Arlman mechanism similar to $Ni^{2+}$ transition metal catalysts. By changing the identity of the main group metal, one can change the resulting product distribution, while, operating with catalytic rates similar to that of $Ni^{2+}$. In addition, $Ga^{3+}$, and $Zn^{2+}$ do not reduce to metallic nanoparticles at high temperatures allowing for higher reaction temperatures. Overall, this study shows exciting potential for applying main group catalytic sites for several different chemistries, i.e., hydrogenation, dehydrogenation, and oligomerization, albeit under different reaction conditions with potential for other catalytic chemistries. Ongoing studies are being conducted to understand the differences in product selectivity and rate on $Zn^{2+}$ and $Ga^{3+}$ metal sites at varying reaction conditions.

## Methods

Single site Ga/SiO₂ and Zn/SiO₂ were prepared following the procedures previously reported in literature, using standard catalyst synthesis techniques, and compared to a Ni/SiO₂ control[22,23,48].

Ga/SiO₂ was synthesized with a chelating agent to prevent the formation of Ga₂O₃ using pH-controlled incipient wetness impregnation (IWI). 10 g of Davisil silica with grade 636 (pore size = 60 Å, surface area = 480 m²/g) was impregnated with an aqueous solution containing 1.5 g of gallium nitrate solution (Ga(NO₃)₃×H₂O, Fluka chemical) and 1.5 g of citric acid (Sigma Aldrich) dissolved in Millipore water. The catalyst was dried for 16 h at 125 °C and then calcined at 500 °C for 3 h. Atomic absorption spectroscopy (AAS) was used to determine that the final catalyst contained ~2.6 wt% Ga.

Zn/SiO₂ was synthesized using pH-controlled strong electrostatic adsorption (SEA). A solution containing 2.5 g of zinc nitrate hexahydrate (Zn(NO₃)₂×6H₂O, Sigma Aldrich) was made and the pH was adjusted to 11 using 30% ammonium hydroxide (NH₄OH) solution, until a clear solution was obtained. 10 g of Davisil silica was suspended 100 mL of Millipore water in a separate beaker and the pH was adjusted to 11 using NH₄OH. The Zn solution was added rapidly to the SiO₂ solution and stirred for 20 min. After the solid was settled, the solution was decanted, and the

resulting slurry was washed with Millipore water and collected by vacuum filtration. The catalyst was dried for 16 h at 125 °C and then calcined at 300 °C for 3 h. AAS was used to determine that the final catalyst contained ~4.0 wt% Zn.

Ni/SiO₂ was prepared by pH-controlled SEA. A solution containing 3.0 g of nickel nitrate hexahydrate (Ni(NO₃)₂×6H₂O) was prepared and the pH was adjusted to 11 using 30% NH₄OH solution until a clear blue solution was obtained. 10 g of Davisil silica was added to the solution and the suspension was stirred for 20 min. At the end of the reaction, additional NH₄OH was added to the solution to maintain a pH of 11. The suspension was stirred for another 10 min before being filtered and the catalyst was recovered. The catalyst was dried for 16 h at 125 °C and then calcined at 300 °C for 3 h. AAS was used to determine that the final catalyst contained ~2.7 wt% Ni.

Oligomerization tests were performed at atmospheric pressure in pure ethylene or pure propylene using a fixed bed reactor of 3/8-inch OD. The weight of catalyst loaded into the reactor ranged from 0.5 to 1 g and was diluted with silica to reach a total of 1 g. The catalyst was treated in 50 ccm of N₂ while it ramped to 250 °C for the reaction. The reaction was performed in 100% $C_2H_4$ using GHSVs ranging from 0.08 to 0.38 s⁻¹. Products from the atmospheric pressure reactor were analyzed with a Hewlett Packard (HP) 6890 Series gas chromatograph (GC) using a flame ionization detector (FID) with an Agilent HP-Al/S column (25 m in length, 0.32 mm ID, and 8 μm film thickness).

High pressure oligomerization was performed in a stainless steel, fixed bed reactor of 1/2-inch OD. 2 g of catalyst was loading into the reactor. The reactor was pressurized to 450 psig (30.6 atm) and the catalyst was treated in 50 ccm of N₂ while it ramped to 250 °C for the reaction. The reaction was performed in a mixture of 10 ccm 5% CH₄/N₂ for an internal standard and 50 ccm 100% $C_2H_4$ at a total pressure of 450 psig. Products were analyzed with a HP 7890 Series GC using a FID with an Agilent HP-1 column (60 m in length, 0.32 mm ID, and 25 μm film thickness).

In-situ XAS was performed at the Ga K (10.3670 keV), Zn K (9.659 keV), and Ni K (8.333 keV) edges at the 10-BM sector at the Advanced Photon Source at Argonne National Laboratory using transmission mode with scan ranges from 250 keV below the edge to 800 keV above the edge. At the Ga K edge, the samples were calibrated to Ga₂O₃ (10.3751 keV). Samples were pressed into a stainless-steel sample holder and placed in a quartz-tube sample cell with gas flow capabilities. The structure of each catalyst was studied after dehydration at 550 °C in He. The sample cell was cooled to room temperature and scanned. The resulting structure of each was compared to known references including Ga acetylacetonate (Ga (AcAc)₃), Ga oxide (Ga₂O₃), Zn oxide (ZnO), and Ni oxide (NiO) to confirm the oxidation state and coordination environment (i.e. coordination number and bond distance). The data was processed using the WinXAS v.3.1 software[56] to find the coordination number and bond distance using standard procedures. Feff6 calculations were performed using Ga₂O₃ (50% at CN = 4, R = 1.83 Å and 50% CN = 6 at 2.00 Å), ZnO (CN = 4, R = 1.98 Å), and NiO (CN = 6, R = 2.09 Å), respectively, for reference. A least-squared fit for the first shell of r-space and isolated q-space were performed on the $k^2$ weighted Fourier transform data over the range of 2.7–10 Å⁻¹ in each spectrum to fit the magnitude and imaginary components.

An understanding of reactive intermediates was obtained on Ga/SiO₂ and Zn/SiO₂ using in situ XAS. A furnace was placed on the beamline around the sample cell to allow for structural measurements at high temperature. Data was continuously collected as the temperature ramped in pure H₂ to 550 °C. Once the structure was stabilized (i.e. the resulting XAS spectra remained unchanged), the cell was cooled to 250 °C in pure H₂ while scanning continuously. When the structure was stabilized, the temperature was held constant at 250 °C and the gas flow was switched from pure H₂ to pure $C_2H_4$. Measurements in He were also obtained at 250 and 550 °C. The XANES were used to determine the oxidation state and geometry while select EXAFS spectra were used to determine the coordination number and bond distances of the M–O bonds (M = Ga, Zn).

To confirm the formation of the metal hydride intermediates and count the number of active metal hydride sites that form, a H₂/D₂ isotopic exchange experiment was performed using a Micromeritics Autochem II 2920 chemisorption analyzer, equipped with a residual gas analyzer (RGA). Calibrations for the H₂, D₂, and HD signal were performed in a bypass line while the sample was being dehydrated at 500 °C in inert gas. For the HD calibration, two separate gas mixtures containing 5% H₂/95% Ar and 5% D₂/95% Ar were combined in different relative amounts in a bypass line to measure initial feed H₂/D₂ ratios in balance Ar compositions. Samples were loaded into a quartz U-tube reactor and treated in flowing air for dehydration at 500 °C before being cooled to 250 °C. The sample was exposed to 5% H₂/Ar for 1 h and then switched to 5% D₂/Ar while the signals for H₂, D₂ and HD were recorded on the RGA. During this time, the H₂ signal returned to its baseline, the D₂ signal increased to its feed value, and the HD signal increased immediately and decreased with time as D₂ reacted with H atoms in metal hydrides to form HD and metal deuterides. Once the HD signal reached baseline values, the gas flow was switched from 5% D₂/Ar to 5% H₂/Ar to quantify the HD formed in the reverse isotopic exchange experiment, and this was repeated for a total of four switches and averaged to estimate the number of metal-hydride sites present.

H₂/D₂ isotopic exchange in a TPSR was performed to identify the number of different metal specific in a catalyst. First, the catalyst was dehydrated at 500 °C treated in air for 2 h. Then, the sample was cooled to 450 °C in air. The catalyst was treated in 5% H₂/95% Ar at 450 °C for 2 h. The temperature was cooled to ambient in 5% H₂/95% Ar. Then, 5% H₂/95% Ar was switched to 5% D₂/95% Ar and the temperature was increased from 35 to 900 °C.

Infrared (IR) spectra were collected using a Nicolet 4700 spectrometer with a Hg–Cd–Te detector (MCT, maintained at $-196\,°C$ by liquid $N_2$). Each spectrum represents the average of 64 scans at $2\,cm^{-1}$ resolution from 4000 to $400\,cm^{-1}$ and were taken using an empty cell background reference ($29\,°C$) collected under dynamic vacuum (rotary vane rough pump, Alcatel 2008A, $<0.01\,kPa$). In a typical experiment, $0.02–0.04\,g\,cm^{-2}$ of sample were pressed into self-supporting wafers of $Ga/SiO_2$ and held in a custom-built quartz IR cell with $CaF_2$ windows. IR cells were inserted into a mineral-insulated heating coil (ARi Industries) contained within an alumina silicate ceramic chamber (Purdue Research Machining Services). The quartz IR cell was connected to a glass vacuum manifold that was used for sample pretreatment and exposure to gas-phase, pure ethylene. When the $Ga/SiO_2$ sample was loaded, it was dehydrated in He at $550\,°C$ for 2 h and a spectrum of the dehydrated sample was obtained. Then, the catalyst was cooled to ambient temperature and exposure to pure $H_2$. The temperature was ramped to $550\,°C$ in pure $H_2$ at a rate of $10\,°C/min$ while spectra were collected every 5 min. The temperature was held at $550\,°C$ in pure $H_2$ for 1 h and then cooled to ambient temperature. The gas was switched to pure $C_2H_4$ and the temperature was ramped at $10\,°C/min$ to $250\,°C$, while collecting spectra every 5 min. The temperature was held at $250\,°C$ for 2 h. A second $Ga/SiO_2$ wafer was prepared and dehydrated using the same method as detailed above. The catalyst was cooled to ambient temperature and exposed directly to pure $C_2H_4$. The temperature was ramped at $10\,°C/min$ to $250\,°$ C while collecting spectra every 5 min, and the $C_2H_4$-treated sample spectra were compared as the sample with and without $H_2$ pretreatment. IR spectra reported here were baseline corrected, and the spectra shown are difference spectra with that of the dehydrated catalyst subtracted from those of the treated catalysts.

$Ga/SiO_2$ structures are based on a recently developed amorphous silica model built using molecular dynamics and continuous dehydration processes[54]. A periodic amorphous silica model ($21.6\,Å \times 21.6\,Å \times 34.5\,Å$; 372 atoms) was used to analyze the energetics of Ga–H formation and ethylene oligomerization. Ga sites were generated by substituting Si atoms and adding a proton to maintain charge balance. All DFT calculations are performed with self-consistent and periodic density functional theory using the Vienna Ab-initio Simulation Package (VASP)[57–61]. The BEEF-VdW exchange-correlation functional[62], using projector-augmented wave (PAW) pseudopotentials[61,63], was employed. A dipole correction was applied parallel to the plane of the slab to reduce image interaction errors. A $k$ point grid of $2 \times 2 \times 1$ was used based on Monkhorst–Pack $k$-sampling. A cutoff energy of 400 eV and a force-convergence criterion of $20\,meV\,Å^{-1}$ for local minima were considered. Transition state geometries were obtained through the climbing-image nudged-elastic-band (NEB) method[64,65], where for each elementary step, seven images were generated as the initial guesses using the Image-Dependent Pair Potential pre-optimizer[66]. After an NEB calculation was converged, where the force exerted on each image was below $100\,meV\,Å^{-1}$, the Lanczos diagonalization approach was employed to locate the transition state with a force convergence criterion of $80\,meV\,Å^{-1}$[67]. The harmonic vibrational states were used for zero-point vibrational energy corrections ($E_{ZPE}$), and these also formed the basis for estimating entropies of the adsorbates. However, for the vibrational modes with low wave numbers ($<150\,cm^{-1}$), particle-in-a-box (PIB) and free rotor schemes were used for calculating their contributions to the entropies, depending on the geometric characteristics of the vibration (see Supporting Information for an example). Free energies, evaluated at $250\,°C$, were obtained using the following equation: $G = E_{DFT} + E_{ZPE} - TS$, where $E_{DFT}$ is the ground-state potential energy calculated using DFT. The calculation of adsorption energy ($G_{ads}$) uses the reference site energy ($G_{Ga}$), which can either be an empty Ga site or Ga hydride, and the gaseous ethylene molecule at 1 atm ($G_{ethylene}$): $G_{ads} = G_A - G_{Ga} - X \times G_{ethylene}$, where $X$ is a stoichiometrically appropriate number of reference ethylene molecules.

## Data availability
The authors confirm that the data in this article and the supplementary information, which support the findings of this study, are available from the authors.

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

## Acknowledgements

This work was supported by the National Science Foundation under Cooperative Agreement no. EEC1647722. G.Z. would like to acknowledge the National Natural Science Foundation of China (21902019) and Fundamental Research Funds for the Central Universities (DUT18RC (3)057 and DUT20RC(5)002). Use of the Center for Nanoscale Materials and Advanced Photon Source, both Office of Science user facilities, was supported by the U.S. Department of Energy, Office of Science, Office of Basic Energy Sciences, under Contract No. DE-AC02-06CH11357. MRCAT operations and the beamline 10-BM were supported by the Department of Energy and the MRCAT member institutions. Information Technology at Purdue (West Lafayette, IN), and computational resources from the National Energy Research Scientific Computing Center is gratefully acknowledged.

## Author contributions

The project was conceived by N.J.L., J.T.M., and G.Z. Catalysts preparation and testing was performed by N.J.L., A.Q., E.E., and R.N. GC–MS spectra on liquid products were collected and analyzed by R.C. In situ XAS measurements and structural analysis were performed by N.J.L. FTIR was performed by J.C.V.-V. under the supervision of R.G. Isotopic exchange experiments were performed by A.S. under the supervision of R.G. Mechanism analysis via DFT calculations was conducted by Y.X. under the supervision of J.G. Equal contributions were made by N.J.L. and Y.X.

## Competing interests

The authors declare no competing interests.
