## [Peer Review File · Nature Communications]

REVIEWER COMMENTS

Reviewer #1 (Remarks to the Author):

The authors have made a few modifications to the manuscript, but I still have two fairly major concerns:

- The Shultz-Flory coefficients were determined based on slopes just a few data points, sometimes as few as 2. Hence, it is not clear whether the coefficients are reliable. For example, the Ga/SiO₂ data on Figure 1d suggests that you could get fairly different slopes from choosing different pairs of points. Perhaps the trends the authors discuss are robust to these uncertainties, but as the manuscript stands this is not clear.
- As stated before, the DFT barriers for some steps are quite high (> 2 eV). It seems the primary goal of the DFT calculations is to establish a plausible mechanism/site, but it's unclear whether plausibility is truly established. The authors still need some justification that these barriers are surmountable, perhaps comparison to other systems from previous work and/or some simple back-of-the-envelope calculations.

Reviewer #2 (Remarks to the Author):

I examined the response of the authors to the comments for the previous submission to [REDACTED]. The explanation is understood in terms of the new insight delivered by this work. However, readers, if accepted, will have an argument on the differences in reaction rate and product selectivity. For this possible issue, the authors would better discuss the limitation and future examination in the manuscript.

On the other hand, the measured gas chromatograms of gas products and GC-MS chromatograms of liquid products need to be provided. This will be very important and valuable for reproducibility and also for future studies on olefin oligomerization with single site catalysts.

Another issue is related with catalyst stability. In the revised conclusion the authors state that Ga³⁺ and Zn²⁺ are readily regenerated once deactivated. However, I cannot find any time-on-stream data if I do not miss. According to the response to one of the comments raised by Reviewer #3, the reaction may seem to proceed over 8 hours. How are reaction rates and product distributions of Ga³⁺ and Zn²⁺ single

site catalysts varied against the reaction time? The plots relevant to catalyst stability should be presented no matter how activity is changed. Additionally, the statement in Conclusion needs to be supported by experimental evidence: therefore, actual results about catalyst regeneration and deactivation would be shown if you have.

Reviewer #1 (Remarks to the Author):

The authors have made a few modifications to the manuscript, but I still have two fairly major concerns:

- The Shultz-Flory coefficients were determined based on slopes just a few data points, sometimes as few as 2. Hence, it is not clear whether the coefficients are reliable. For example, the Ga/SiO₂ data on Figure 1d suggests that you could get fairly different slopes from choosing different pairs of points. Perhaps the trends the authors discuss are robust to these uncertainties, but as the manuscript stands this is not clear.

Thank you for this comment, and we agree that there are too few data points to accurately determine the precise SF coefficients. This is in part due to the low conversion at low pressure and small concentration of higher molecular weight products under these conditions. Unfortunately, at higher pressure our reactor can't quantitatively collect all products simultaneously. Products greater than about C₆ partially condense, thus are not quantitatively analyzed. Our reactor is not a large enough scale to collect all liquid products for an accurate determination. At this stage, the best we can do is get approximate values, which show there are differences between Zn²⁺ and Ga³⁺ with higher molecular weight products from the latter. We have collected the liquid products from the high pressure and higher conversion reactions, and as the gas analysis indicates, Ga does make higher molecular weight products. The GC-MS results, requested by Reviewer 2, with peak identification was added to the SI in Figure S3. Further studies to optimize the catalysts, process conditions and quantitatively identify the products are in progress.

On page 7 we have revised the discussion of the SF coefficients and indicated that these are based on a limited number of data points and are approximate values, but do indicate that Ga³⁺ lead to higher MW products than Zn²⁺.

- As stated before, the DFT barriers for some steps are quite high (> 2 eV). It seems the primary goal of the DFT calculations is to establish a plausible mechanism/site, but it's unclear whether plausibility is truly established. The authors still need some justification that these barriers are surmountable, perhaps comparison to other systems from previous work and/or some simple back-of-the-envelope calculations.

We agree with the reviewer that the barriers are high. To address this issue, we have undertaken a larger survey of Ga³⁺ sites and have recalculated energies on one promising candidate, as well as updating all DFT-related figures in both the main text and SI. The site is less constrained than our previous model, but it is still four-coordinated, which is consistent with our experimental results. On this modified Ga³⁺ site, we have observed decreased barriers (by approximately 0.5 eV) for all elementary steps. Hence, we observe that a less-constrained Ga³⁺ site can contribute to a much more favorable energy landscape. Similar site heterogeneity has been observed in the ethylene polymerization study using Cr³⁺ sites, as well as propane dehydrogenation using Ga³⁺, where similar amorphous silica models and site creation schemes were used, suggesting that any such less constrained sites could contribute to the reactivity (Praveen, C. S. *et al.*, (2020)

ChemRxiv; Floryan, L. *et al.*, (2017) Journal of Catalysis). Additional insights into how these barriers quantitatively affect the reaction rates could, in principle, be obtained by microkinetic modeling, but those analyses are currently beyond the scope of the present manuscript.

Reviewer #2 (Remarks to the Author):

I examined the response of the authors to the comments for the previous submission to [REDACTED]. The explanation is understood in terms of the new insight delivered by this work. However, readers, if accepted, will have an argument on the differences in reaction rate and product selectivity. For this possible issue, the authors would better discuss the limitation and future examination in the manuscript.

This work aims to demonstrate the potential of main group metal ions as catalysts for oligomerization and the related hydrogenation/dehydrogenation reactions, which appear to have similar intermediates and elementary steps. The rates have been determined based on 1) total metal loadings, and 2) estimated fractions of metal that have formed hydrides under the reaction conditions. Based on the total metal loading, the rates differ by about 3 times. Correcting for the steady state number of M-H sites, the rates are similar. Since the steady state number of sites in these catalysts seems to be dependent on the catalyst composition, reaction temperature and perhaps even the operating pressures, clearly improved TORs will be possible once one can determine the fraction of active sites under operando conditions. This is beyond the scope of the current study.

As discussed above for reviewer 1, our small-scale, high pressure reactors are not designed to produce large volumes of liquid products for thorough identification of all products. However, as indicated by the gas analysis, at higher conversions at higher pressure Ga³⁺ does lead to higher molecular weight (liquid) products than Zn²⁺. Currently, we do not understand why Ga³⁺ leads to higher molecular weight products compared to Zn²⁺. Thus, there are future opportunities to determine the fundamental understandings about these differences. The current results, however, suggest the catalyst can change the molecular weight distribution, and we expect that future studies may lead to understanding of which mechanistic step controls this and how the catalyst may be optimized to control the products. We are working on these issues currently and will report on these in the future.

On the other hand, the measured gas chromatograms of gas products and GC-MS chromatograms of liquid products need to be provided. This will be very important and valuable for reproducibility and also for future studies on olefin oligomerization with single site catalysts.

The total chromatogram for GC-MS results with peak identification was added to the SI in Figure S3. This shows the formation of linear hydrocarbons up to at least C15 and the formation of trace amounts of aromatics. Small amounts of higher molecular weight products did not elude from the column and could not be identified.

An another issue is related with catalyst stability. In the revised conclusion the authors state that Ga³⁺ and Zn²⁺ are readily regenerated once deactivated. However, I cannot find any time-on-stream data if I do not miss. According to the response to one of the comments raised by Reviewer #3, the reaction may seem to proceed over 8 hours. How are reaction rates and product distributions of Ga³⁺ and Zn²⁺ single site catalysts varied against the reaction time? The plots relevant to catalyst stability should be presented no matter how activity is changed. Additionally, the statement in Conclusion needs to be supported by experimental evidence: therefore, actual results about catalyst regeneration and deactivation would be shown if you have.

We have provided a plot showing stable performance in the SI, Figure S2. Prior to each reaction, both the Zn²⁺ and Ga³⁺ catalysts were white in color. After reaction, the catalysts turned beige or brown. The spent catalysts were calcined at 500°C in flowing air for 3 hours, which restored the catalysts to their original white color and catalyst performance. The regenerability of these materials, while interesting and important for practical applications, is beyond the scope of the present manuscript and the claim in the conclusion has been removed.

REVIEWERS' COMMENTS

Reviewer #1 (Remarks to the Author):

The authors have addressed my concerns to an acceptable degree.

Reviewer #2 (Remarks to the Author):

The manuscript is well revised by reflecting the response to the previous comments. Therefore, I strongly consider that the present version is worth being published in Nature Communications.

Nevertheless, there is one thing to be added into the manuscript: that is, compositions of linear alpha olefins such as 1-hexene and 1-octene. As the authors know well, these olefins have received particular interests in industry because they are commonly used as co-monomer for polyethylene production. In this respect, the authors made a few discussions: equilibrium distributions of butenes and higher molecular weight products. I think a little more quantitative results be better for potential readers. It is recommended that the fractions of 1-butene in C4=, 1-hexene in C6=, and 1-octene in C8= are added into the manuscript.

Response to Reviewers and Editor
Final Changes to Manuscript: *Nature Communications*, NCOMMS-20-35524-T

Reviewer 1

The authors have addressed my concerns to an acceptable degree.

Reviewer 2

The manuscript is well revised by reflecting the response to the previous comments. Therefore, I strongly consider that the present version is worth being published in *Nature Communications*.

Nevertheless, there is one thing to be added into the manuscript: that is, compositions of linear alpha olefins such as 1-hexene and 1-octene. As the authors know well, these olefins have received particular interests in industry because they are commonly used as co-monomer for polyethylene production. In this respect, the authors made a few discussions: equilibrium distributions of butenes and higher molecular weight products. I think a little more quantitative results be better for potential readers. It is recommended that the fractions of 1-butene in C4=, 1-hexene in C6=, and 1-octene in C8= are added into the manuscript.

Response: We agree with this comment. Commercial oligomerization processes operate to produce linear, alpha-olefins for use as co-monomers for ethylene polymerization. Olefin isomerization or branching is an undesirable selectivity. Our catalysts make a mixture of linear olefins, which are not suitable for olefin polymerization applications; however, these have a MW that is highly attractive for hydrogenation to diesel fuel applications. We have added additional text to clarify that there is a mixture of olefin isomers for C6 and C8 (and other products) as requested by the reviewer.

Changes to manuscript (page 6):

For each catalyst, there were many isomers of linear olefins. For example, there were five C₆ isomer products representing 1-hexene cis and trans- 2- hexene, and cis and trans – 3- hexene. Similarly, there were many isomer products for the C₈ and other products.